

# Histopathologic evaluation of postmortem autolytic changes in bluegill (*Lepomis macrohirus*) and crappie (*Pomoxis anularis*) at varied time intervals and storage temperatures

Jami George[1], Arnaud J. Van Wettere[2], Blayk B. Michaels[3], Debbi Crain[3] and Gregory A. Lewbart[1]

[1] Clinical Sciences, North Carolina State University College of Veterinary Medicine, Raleigh, NC, United States
[2] Department of Animal, Dairy & Veterinary Sciences, Utah State University, Logan, UT, United States
[3] Live Exhibits, Bass Pro Shops, Springfield, MO, United States

## ABSTRACT

Information is lacking on preserving fish carcasses to minimize postmortem autolysis artifacts when a necropsy cannot be performed immediately. The purpose of this study was to qualitatively identify and score histologic postmortem changes in two species of freshwater fish (bluegill—*Lepomis macrochirus*; crappie—*Pomoxis annularis*), at varied time intervals and storage temperatures, to assess the histologic quality of collected samples. A pooled sample of 36 mix sex individuals of healthy bluegill and crappie were euthanized, stored either at room temperature, refrigerated at 4 °C, or frozen at −20 °C, and then necropsied at 0, 4, 24, and 48 h intervals. Histologic specimens were evaluated by light microscopy. Data showed that immediate harvesting of fresh samples provides the best quality and refrigeration would be the preferred method of storage if sample collection had to be delayed for up to 24 h. When sample collection must be delayed more than 24 h, the preferred method of storage to minimize autolysis artifacts is freezing if evaluation of the gastrointestinal tract is most important, or refrigeration if gill histology is most important. The gill arch, intestinal tract, followed by the liver and kidney were the most sensitive organs to autolysis.

## INTRODUCTION

When death occurs, cellular metabolic activity declines at varying rates, while enzymes proceed unchecked. Autolysis results in cellular breakdown with consequences of structural changes that often distort the line between natural and pathological processes (*Carson & Hladik, 2009*; *Cooper, 2012*; *Dettmeyer, 2011*; *Elmore, 2007*; *Roe, Gartrell & Hunter, 2012*; *Sterne, Titley & Christie, 2000*; *Tavichakorntrakool et al., 2008*; *Zdravković, Kostov & Stojanović, 2006*). Postmortem samples are frequently utilized in clinical studies, forensic investigations, and biomedical research (*Tavichakorntrakool et al., 2008*). Of the numerous available diagnostic techniques employed, histology is commonplace and offers detection of

Corresponding author
Gregory A. Lewbart,
greg_lewbart@ncsu.edu

subtle morphological tissue changes early in disease processes not easily recognized grossly (*Heil, 2009*). Common knowledge dictates that the shorter the time between death and tissue fixation for histopathologic evaluation, the better tissue preservation will be, leading to more accurate results (*Tavichakorntrakool et al., 2008*). For various reasons postmortem sample collection is often delayed. Presently there is a lack of information regarding how to best preserve fish carcasses to minimize autolysis until trained personnel can process them for histopathology.

Past investigations have looked at histological findings following prolonged postmortem interval (PMI) in mammals (*Dettmeyer, 2011*; *Erlandsson & Munro, 2007*). The detection of usable microscopic findings in relation to the PMI "is by nature temporary and dependent" due to a multitude of factors (*Dettmeyer, 2011*; *Erlandsson & Munro, 2007*). When considering fish, the factors include, but are not limited to: ectothermy, body size, fat content, management in captivity, handling, habitat, life stage, health status, amount of ingesta, bacterial flora, medications, time, and type of storage (*Chow & Zhang, 2011*; *Cooper, 2012*; *Dettmeyer, 2011*; *Heil, 2009*; *Mukundan, Antony & Nair, 1986*; *Roe, Gartrell & Hunter, 2012*; *Sterne, Titley & Christie, 2000*; *Tavichakorntrakool et al., 2008*; *Tomita et al., 2004*; *Zdravković, Kostov & Stojanović, 2006*). Organs from dead fish are known to autolyze more quickly than mammalian organs and will often mask ante-mortem changes (*Ferguson, 2006*; *Heil, 2009*; *Mukundan, Antony & Nair, 1986*; *Roberts, 2012*). One possible explanation is that fish cells rapidly utilize glucose and glycogen from the stress of handling and/or shipping which results in increased lactic acid, a decreased pH, and exacerbation of autolysis at the time of death (*Gatica et al., 2008*; *Mukundan, Antony & Nair, 1986*). Improper temperature storage ($-1$ to $-4\,^{\circ}$C) worsens autolysis due to an enhanced rate of glycolysis (*Mukundan, Antony & Nair, 1986*; *Thomas, Pankhurst & Bremner, 1999*).

Accurate interpretation of gross and microscopic lesions at postmortem relies heavily on good sample preservation and is impeded by autolysis that follows a prolonged PMI (*Roe, Gartrell & Hunter, 2012*). In the case of wild, farm-raised, or ornamental fishes, there is frequently a long distance between the site of death and a diagnostic facility. Therefore, postmortem examinations are often performed on frozen-thawed, desiccated, and severely autolyzed fishes (*Roe, Gartrell & Hunter, 2012*). The effects of freezing organs have been well studied and consists of cell membrane lysis, loss of stain uptake, eosinophilic extracellular fluid accumulation, cell shrinkage, fractures, disruption of blood vessel walls, and development of pseudo-lesions (*Baraibar & Schoning, 1985*; *Chow & Zhang, 2011*; *Roe, Gartrell & Hunter, 2012*; *Sterne, Titley & Christie, 2000*). These artifacts are compounded by autolysis and make interpretation extremely challenging. It is therefore crucial to determine how best to preserve fish carcasses to minimize tissue autolysis and artifacts in order to improve our ability to define changes, diagnose diseases and ultimately better the health and welfare of fish stocks (wild and farmed). The purpose of this study was to qualitatively identify and score histologic postmortem changes in two related species of freshwater fish, bluegill (*L. macrochirus*) and crappie (*P. annularis*), at varied time intervals and storage temperatures, to assess the histological quality of collected samples.

**Table 1  A summary of the experimental design including the following: time of collection, storage method, and temperature of the room and fish.** All were performed prior to harvesting each sample.

| Time of collection | Storage method | | | Fish body temperature | Room temperature |
|---|---|---|---|---|---|
| Hours | Room temperature | Refrigerated | Frozen | Celsius | Celsius |
| 0 | X | | | 20.20 | 20 |
| | X | | | 18.40 | 20 |
| 4 | | X | | 13.40 | 20 |
| | | | X | 9.88 | 20 |
| | X | | | 18.70 | 20 |
| 24 | | X | | 3.10 | 20 |
| | | | X | 1.72 | 20 |
| | X | | | 18.20 | 18.89 |
| 48 | | X | | 3.44 | 18.89 |
| | | | X | −0.11 | 18.89 |

## MATERIALS AND METHODS

### Subjects

A total of 36 bluegill (*L. macrochirus*) and crappie (*P. annularis*) scheduled for depopulation at a large commercial retail facility were utilized. These closely related species are members of the Centrarchidae. They averaged 340 g and both sexes were randomly included. None of the fish were noted to display clinical or gross signs of disease.

### Experimental design

All 36 fish were euthanized by means of tricaine methanesulfonate (MS-222) overdose at 215 ppm, separated into three groups of 10 (mixed species), and then individually double bagged in plastic. The fish remained in bags until processing. Afterward, nine were kept at room temperature (20 °C), nine were refrigerated (4 °C), and nine were frozen (−20 °C). Necropsies were performed on the remaining three fish from each cohort, deemed the control groups, in which organs were immediately harvested (also known as procedural starting time or the 0 h mark). Four hours later, organs were harvested from three fish at room temperature and three refrigerated fish, while three fish were removed from the freezer, set to thaw in cool water for 1 h, and then harvested. This routine was repeated at 24, and 48 h intervals in which all fish were sampled. Room temperature and internal body temperatures were recorded at the time of collection (Table 1). Samples from all fish were harvested in the following order: gills, upper and lower intestines, spleen, liver, stomach, gonad, posterior kidney, anterior kidney, heart, skin with underlying muscle, and brain. All samples were placed in sealed containers of 10% neutral buffered formalin (Sigma-Aldrich, St. Louis, Missouri, USA). Every effort was made to handle fish consistently and promptly in order to minimize the impact of minute-level time differences.

### Histologic evaluation

The following fixed samples were trimmed, routinely processed for histopathology, and stained with hematoxylin and eosin: gill arch, skin with underlying skeletal muscles and

scales, kidney, liver, spleen, gonads, heart, stomach, pyloric caeca, intestines, and brain. Gill arch and skin with underlying skeletal muscle were decalcified with 10% formic acid for 24 h prior to processing. Histology evaluation was performed using a Nikon Eclipse 50i microscope (Nikon Instruments Inc, Melville, NY, USA). Samples were evaluated and scored according to the extent of autolysis in the section examined using a semi-quantitative scale compared to the control samples: minimal autolysis (<5%), mild autolysis (5–10%), moderate autolysis (10–50%), and severe autolysis (>50%).

## Criteria of autolysis

Postmortem changes include both autolysis and putrefaction (bacterial breakdown of organs). The definition of autolysis is the self-digestion or degradation of cells by hydrolytic enzymes normally present within cells postmortem (*Carson & Hladik, 2009*; *Mukundan, Antony & Nair, 1986*; *Zdravković, Kostov & Stojanović, 2006*). Criteria for autolysis comprised the following factors. The overall severity of autolysis was based on the number of factors present and the magnitude of change in the affected cells:

(1) Pyknosis
(2) Karyorrhexis
(3) Karyolysis
(4) The absence of a nucleus due to complete dissolution or lysis
(5) Cellular edema/swelling
(6) Failure to take up stain
(7) Intracytoplasmic vacuolation
(8) Putrefaction
(9) Altered architecture of tissue unrelated to a pathological process

## RESULTS

All fish and room temperatures just prior to necropsy are listed in Table 1. Autolysis scores for each sample are shown in Table 2.

### Fresh samples (control group) (Fig. 1)

No significant architectural and cellular histologic changes were observed and stain uptake was optimal for all organs. In the gill, mild artifactual secondary lamellar epithelial lifting was the most common finding. Incidental lesions present included few focal aneurisms of the lamellae, mild multifocal lamellar fusion, and secondary hyperplasia of lamellar epithelial (pavement) cells at interlamellar sulci were also present.

Within the skin and muscle, artifactual separation of the cuticle and epidermis was present, with minimal lifting of the basement membrane and stratus spongiosum of the dermis. The brain had mild artifactual lifting of the dura matter and mild separation of the stratum periventriculare from the tegmentum. The gastrointestinal villi were present and intact. Within the intestines, focal areas of minimal artifactual lifting of the lamina propria from the intestinal epithelium were appreciated.

George et al. (2016), *PeerJ*, DOI 10.7717/peerj.1943

Peerj

**Table 2** **Autolysis scores[a] for fresh, refrigerated, room temperature, and frozen tissue samples taken at 0, 4, 24, and 48 h.[b]**

| Samples | Gill arch | Skin | Skeletal muscle | Kidney | Liver/pancreas | Spleen | Gonads | Heart | GI | Brain |
|---------|-----------|------|-----------------|--------|----------------|--------|--------|-------|-----|-------|
| 0 h Fresh | – | – | – | Minimal | Minimal | – | – | – | – | – |
| 4 h Ref. | Minimal | Minimal | Minimal | Minimal | Minimal | Minimal | Minimal | Minimal | Minimal | Minimal |
| 4 h Z | Mild | Minimal | Mild to moderate | Mild | Minimal | Minimal | Minimal | Minimal | Mild | Mild |
| 4 h RT | Mild to moderate | Minimal | Minimal | Mild | Minimal | Minimal | Minimal | Minimal | Mild | Mild |
| 24 h Ref. | Moderate | Minimal | Minimal | Mild | Mild | Mild | Mild | Mild | Mild | Mild |
| 24 h Z | Severe | Moderate | Mild to moderate | Mild to moderate | Mild to moderate | Mild to moderate | Mild | Mild | Mild to moderate | Mild to moderate |
| 24 h RT | Severe | Moderate | Mild to moderate | Moderate | Moderate | Moderate | Moderate | Moderate | Moderate to severe | Moderate to severe |
| 48 h Ref. | Moderate to severe | Moderate | Mild to moderate | Moderate | Moderate | Mild to moderate | Mild to moderate | Mild to moderate | Severe | Mild to moderate |
| 48 h RT | Severe | Severe | Moderate | Severe | Severe | Severe | Severe | Severe | Severe | Severe |
| 48 h Z | Severe | Severe | Mild to moderate | Moderate | Moderate to severe | Moderate | Moderate | Mild to moderate | Mild to moderate | Mild to moderate |

**Notes.**
[a]Minimal (<5%); Mild (5–10%); Moderate (10–50%); Severe (>50%).
[b]Ref., refrigerated; Z, frozen; RT, room temperature; h, hour(s); GI, gastro-intestinal.

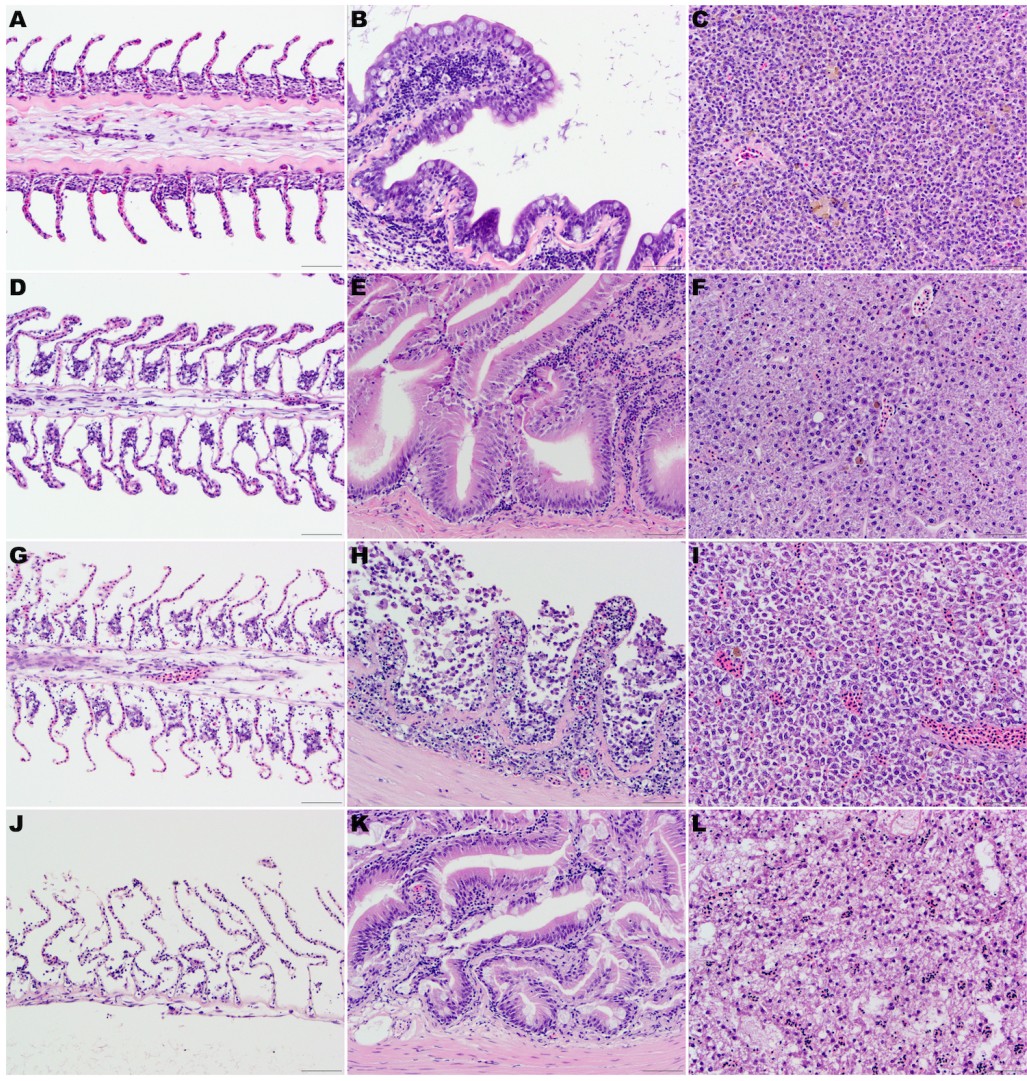

**Figure 1** **(A, D, G) and (J) Gill lamellae, formalin-fixed at the time of necropsy (A), after 24 h at 4 °C (D), after 48 h at 4 °C (G), and thaw 48 h after being frozen at the time of euthanasia (J).** Note the normal filament covered by a closely apposed layer of flat lamellar epithelial cells in the sample formalin fixed at the time of euthanasia (A). After 24 h (D) or 48 h (G) at 4 °C, note the widespread epithelial capillary separation, loss of lamellar epithelial cells and dilated extracellular spaces in the filament. These changes are most pronounced in the fish that were frozen at the time of euthanasia and then thawed in water. (B, E, H) and (K) Intestine, formalin-fixed at the time of necropsy (B), after 24 h at 4 °C (E), after 48 h at 4 °C (H), and thawed 48 h after being frozen at the time of euthanasia (K). Note the well-preserved epithelial cells in the sample formalin fixed at the time of euthanasia (B) or after 24 h (E) at 4 °C. After 48 h (H) at 4 °C, there is marked autolysis with loss of cellular details, epithelial cells and architecture. Mild widespread epithelial separation and loss of cellular details and dilated extracellular spaces were present in the fish that were frozen at the time of euthanasia and then thawed in water (K). (C, F, I, L) Liver, formalin-fixed at the time of necropsy (C), after 24 h at 4 °C (F), after 48 h at 4 °C (I) and thawed 48 h after being frozen at the time of euthanasia (L). The cellular details and architecture are well-preserved in the samples formalin fixed at the time of euthanasia (C) or mildly decreased in the samples fixed in formalin after 24 h (F) at 4 °C. After 48 h (I) at 4 °C, moderate loss of cellular details and widespread cell separation are evident. Loss of cellular details and dilated extracellular spaces were pronounced in the fish that were frozen at the time of euthanasia and then thawed in water (L). Hematoxylin-eosin stain, 400×, bar = 50 µm.

## Samples at 4 h (Fig. 1)

All samples had optimal stain uptake and the histological architecture and cellular preservation were good to moderate. Autolytic changes were minimal and it was not possible to distinguish between the skin, liver, pancreas, spleen, gonads, and heart of fish kept refrigerated, frozen for 4 h or at room temperature. Minimal secondary lamellar epithelial lifting, loss of cellular details at the lamellae tips and mild vacuolation with increase clear spaces between epithelial and basal cell layers of lamellae were present in the refrigerated fish. Similar histologic changes but more pronounced were present in the fish kept frozen and at room temperature. Rare secondary lamellar aneurysms were also present in all groups.

Minimal to mild amount of clear spaces were present between myocytes in all groups. Mild to moderate linear fractures were seen between myocytes in the fish kept frozen.

In the kidney, a mild amount of clear spaces were seen between tubules and hematopoietic cells as well as mild autolysis especially in hematopoietic cells were present in all groups.

Mild lifting of the dura mater and mild separation of the stratum periventriculare from the tegmentum was present in all groups. Mild increased clear spaces were seen between the stratum fibrosum marginale and the stratum opticum.

Small amount of clear spaces separating epithelial cells and rare karyolysis and pyknosis were present in the intestinal mucosa of all fish groups. Minimal to mild increased clear spaces in the intestinal lamina propria and inner circular smooth muscle were present.

Testicular samples were lacking in room temperature samples.

## Samples at 24 h (Fig. 1)

All samples had good stain uptake and the histological architecture and cellular preservation were good to poor. With the exception of the gill, autolytic changes were minimal or mild in the fish kept at 4 C for 24 h. In the gill, moderate breakdown of lamellae with epithelial lifting, moderate vacuolization, loss of cellular details and condensation and fragmentation of the nuclei were present.

Autolytic changes were mild to moderate in the frozen samples with the exception of the gill. Severe autolysis and dissolution of the gill arch epithelium could be seen, as well as more than 50% loss of lamellar histologic architecture.

Samples at room temperature had moderate to severe autolytic changes, especially prominent in the gill, gastrointestinal tract, and brain. Moderate to severe dissolution of the gill arch epithelium and lamellar histological architecture could be seen, along with the loss of lamellae structure.

In the skin and musculature, minimal autolytic changes were observed in the fish kept refrigerated for 24 h.

In the fish frozen and kept at room temperature, moderate autolysis and loss of the epithelium and mild to moderate freezing artifacts of the musculature could be found in all samples.

In the liver, pancreas, spleen and kidney, autolytic changes were mild in refrigerated fish and moderate in the fish kept at room temperature for 24 h. In the frozen samples, autolysis

was mild to moderate with occasional fracture artifact seen. Interestingly, changes in the pancreases were mild to moderate, even in the fish kept at room temperature; vacuolation and cellular separation by clear spaces could be seen.

Autolytic changes in the testes and heart were mild in the refrigerated and frozen fish but more pronounced in the fish kept at room temperature. They consisted of mild to moderate nuclear autolysis. Occasional fractures were seen in the frozen samples. In the heart samples kept at room temperature, postmortem bacilli were observed in the blood and cardiac tissue. Deterioration was commonly seen and putrefaction was found in 2/3 of samples. No ovaries were present in the fish collected at the 24 h time point.

While autolytic changes were mild in the brain and gastrointestinal tract of the fish kept refrigerated, or mild to moderate in the fish kept frozen, autolytic changes were much more pronounced in the fish kept at room temperature. Intestinal villi architecture was maintained, with mild congealing and vacuolation of the epithelium, in refrigerated and frozen samples. Mild to moderate vacuolation and moderate autolysis were seen in the lamina propria and inner circular smooth muscle. Changes were more pronounced in the samples kept at room temperature and postmortem bacterial bacilli were present.

### Samples at 48 h (Fig. 1)

All samples had a mild to moderate decrease in stain uptake and the histological architecture and cellular preservation were mild to poor.

Gill arch and skin were moderately to severely autolyzed in all groups. Gills had an empty skeleton-like appearance, with moderate to severe dissolution of the gill arch epithelium and lamellar histological architecture, loss of lamellae structure, and bony destruction and putrefaction. Most of the epithelium was lost in the skin samples. Muscular autolysis was moderate in the fish kept at room temperature and less pronounced in the refrigerated and frozen samples. Mild to moderate freezing artifacts were present in the frozen fish. Putrefaction of the muscle kept at room temperature was evident. The liver, kidney and pancreas kept refrigerated or frozen showed moderate autolysis maintaining histologic structural integrity but with loss of cellular details. Autolysis was severe in the liver, kidney, and pancreas kept at room temperature. The spleen, gonad, heart, and brain were best preserved in the fish kept refrigerated with only mild to moderate autolysis present. Autolysis was most pronounced, severe, in the fish kept at room temperature.

Autolysis of the gastrointestinal tract was least pronounced in the fish kept frozen with moderate preservation of the epithelium; enterocytes brush border was present multifocally. Autolysis and putrefaction of the gastrointestinal tract of the fish kept refrigerated or at room temperature was severe.

## DISCUSSION

As expected, immediate fixation of the organs provided best preservation for histology, while most prominent autolysis occurred in samples kept for 48 h at room temperature. These results show that when immediate sampling and fixation is not possible, but will be performed within 24 h, the ideal storage method would be refrigeration. If necropsy must be delayed more than 24 h, refrigeration is the best preservation method, at least up to

48 h, for gill tissue. Interestingly, if histology of the gastrointestinal tract is sought, freezing is superior to refrigeration. Despite the above findings, we recommend that whenever possible, fresh tissues are fixed immediately in an appropriate fixative, and that for some organs 10% neutral buffered formalin is not the best choice.

Prior research has verified in humans and mammals that the shorter the period is from the time of death to organ fixation, the more accurate acquired data will be (*Mukundan, Antony & Nair, 1986*; *Roe, Gartrell & Hunter, 2012*; *Sterne, Titley & Christie, 2000*; *Tavichakorntrakool et al., 2008*). Our results support this fact for freshwater fish as well. It is known that drastic temperature changes can hasten autolysis (*Mukundan, Antony & Nair, 1986*). Temperatures lower than −1 °C can enhance glycolysis and thus autolysis (*Mukundan, Antony & Nair, 1986*). Studies have also shown that elevated temperature can accelerate autolysis (*Carson & Hladik, 2009*; *Sterne, Titley & Christie, 2000*). Our study found less pronounced autolysis and fewer artifacts develop with refrigeration, as compared to holding the carcass at room temperature, or freezing. Our results are consistent with previous studies on other taxa (*Carson & Hladik, 2009*; *Mukundan, Antony & Nair, 1986*; *Roe, Gartrell & Hunter, 2012*; *Sterne, Titley & Christie, 2000*).

Our results are similar to other studies showing that refrigeration will produce fewer artifacts than freezing regardless of taxa (*Baraibar & Schoning, 1985*; *Mukundan, Antony & Nair, 1986*; *Roe, Gartrell & Hunter, 2012*; *Sterne, Titley & Christie, 2000*). Freeze-thaw effects have been shown to induce cell membrane lysis, fluid shifts into the extracellular space, and disruption of blood vessel walls resulting in blood cell extravasation (*Roe, Gartrell & Hunter, 2012*). Ice crystal formation produces the spaces seen on histology.

Current thoughts state that the onset of autolysis is rapid and more severe in organs that are rich in hydrolytic enzymes such as the pancreas (*Carson & Hladik, 2009*; *Dettmeyer, 2011*). In a suitable acidic environment (low pH), these enzymes would digest organs around their site of origin, causing early-on autolytic changes (*Carson & Hladik, 2009*; *Dettmeyer, 2011*). Other research shows the pancreas, with its high content of hydrolytic enzymes, was relatively resistant to postmortem change up to 24 h after death (*Jones & Trump, 1975*; *Tomita et al., 2004*). One study stated that shrinkage of the acinar cells in the pancreas was not appreciated until 10 h postmortem (*Tomita et al., 2004*). In our study, autolytic changes in the pancreas remained mild to moderate in most fish samples until 24–48 h had elapsed, except for the fish kept at room temperature for 48 h. This supports the belief that the presence of hydrolytic enzymes does not necessarily dictate the onset of autolysis, and cell membrane integrity should be a more important consideration (*Ilse et al., 1979*).

The gill arch and gastrointestinal tracts showed autolytic changes first, followed by kidney, liver and brain, then spleen, gonad, and heart. Skin, and skeletal muscle appeared more resistant to autolysis. It has been established that the rate of autolysis is not the same among different cell types, and is possibly affected by the functional state of the cell (*Ilse et al., 1979*). Autolysis is also less rapid in organs with elastic fibers/collagen (*Carson & Hladik, 2009*; *Dettmeyer, 2011*). Skeletal muscle has been shown to have the greatest delay in postmortem change in rats as compared to the kidney, liver, and heart (*Tomita et al., 2004*). Muscle does not provide a strong acidic environment after death for autolysis to

occur rapidly in (*Takeichi et al., 1984*). In another study, human skeletal muscle histology remained unchanged until 24 h at 4 °C, and 6 h at 25 °C (*Tavichakorntrakool et al., 2008*).

The gill arch is a dominant site of gas exchange, osmoregulation, acid–base balance, and excretion of nitrogenous waste (*Evans, Piermarini & Choe, 2005*). It holds a central role in a fish's physiological response to environmental and internal changes, as well as the principal site of body fluid pH regulation (*Evans, Piermarini & Choe, 2005*). As expected, the gill epithelium exhibited more autolytic change, as opposed to connective tissue, and bone.

When comparing the degree of autolysis of the gill epithelium and filaments in refrigerated samples versus frozen samples, autolysis was more pronounced in frozen tissues. Artifacts created through freeze-thaw procedures induced greater alterations in the histological structure, which was often associated with more autolytic criteria. Thawing of the frozen fish in water may have increased the autolysis change in the gills. The effect on gill histology of thawing in air verses water would be interesting to know.

The results of this study are limited to centrarchid fishes with an average bodyweight of 340 g. Extrapolating to other piscine species, or centrarchids of different sizes, should be undertaken with care, and an understanding that not all fish samples would behave similarly. Unfortunately, for logistical reasons, exact weights for each fish in this study were not recorded. Furthermore, the two species were treated as a population, and not as separate taxa. This limits the study and we recommend that future work in this area more clearly define these criteria.

## CONCLUSIONS

The dynamics involved with postmortem autolysis are numerous. Veterinary pathologists rely heavily on experience and knowledge to determine the cause of death of a variety of species and circumstances, which can often reduce interpretations to being subjective. Compounding this challenge includes unknown PMIs, poor quality sampling, and suboptimal storage. Having an understanding of the longest PMI for a species, which can provide the most reliable data without hindering analyses, as well as the best postmortem storage method, can offer more accurate interpretations of lesions.

Our observations show that organs collected and fixed immediately after death results in optimal preservation for histology of freshwater fish. If sample collection must be delayed for up to 24 h refrigeration is recommended. Results also suggest that if histology of the gill is most important, refrigeration is the best preservation method at least up to 48 h. However, if histology of the gastrointestinal tract is sought, freezing is superior to refrigeration if sample collection must be delayed more than 24 h. The first organs showing autolysis were the gill arch and gastrointestinal tract (stomach, pyloric caeca, intestines). The second group of organs affected by autolysis was the kidney, liver and brain. The third tier included the gonad, heart, and spleen. Skin and skeletal muscle were the most resistant organs to autolysis.

This study is limited by the fact that all specimens were closely related and of similar size. It is possible that different species, fish of smaller or larger size, or isolated organs might not behave similarly. Studies using different species, or fish of varying sizes, are warranted and would help validate the results of our work.

## ACKNOWLEDGEMENTS

We would like to thank Ruth Francis-Floyd, Denise Petty, Derek Bossi, Emily Griffith and Kent Passingham for their efforts and input on this project and the NCSU-CVM pathology and histology staff for their assistance in interpretation and processing of samples.

### Funding

The authors received no funding for this work.

### Competing Interests

Blayk Michaels and Debbi Crain are employed by Bass Pro Shops Inc.

### Author Contributions

- Jami George conceived and designed the experiments, analyzed the data, wrote the paper, prepared figures and/or tables, reviewed drafts of the paper.
- Arnaud J. Van Wettere analyzed the data, contributed reagents/materials/analysis tools, wrote the paper, prepared figures and/or tables, reviewed drafts of the paper.
- Blayk B. Michaels conceived and designed the experiments, performed the experiments, contributed reagents/materials/analysis tools, reviewed drafts of the paper.
- Debbi Crain conceived and designed the experiments, performed the experiments, reviewed drafts of the paper.
- Gregory A. Lewbart conceived and designed the experiments, contributed reagents/materials/analysis tools, reviewed drafts of the paper.

### Animal Ethics

The following information was supplied relating to ethical approvals (i.e., approving body and any reference numbers):

Not applicable. All work was performed on deceased animals.

### Data Availability

The raw data has been supplied as Supplemental Information.

### Supplemental Information

Supplemental information for this article can be found online at http://dx.doi.org/10.7717/peerj.1943#supplemental-information.

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
