# Peer review of "Histopathologic evaluation of postmortem autolytic changes in bluegill (Lepomis macrohirus) and crappie (Pomoxis anularis) at varied time intervals and storage temperatures"

_PeerJ, doi:10.7717/peerj.1943_

## Round 0.1 · original submission · Minor Revisions

· Academic Editor

Minor Revisions

This is an interesting manuscript. As you can see there are a number of comments that need to be addressed. In particular please make sure you are using correct terminology, there are quite a few very useful suggestions in the comments from the reviewers. One of mistakes commonly made and also present in your manuscript is to use the word "tissue" instead of "organ" or "sample". Please check the definition of "tissue" and "organ" and please make sure you are using correct word. Sometimes it is best just to deleted it (for example "tissue samples" should be just "samples") or replaced with "samples" for example "Samples were collected" is better than "Tissues were collected". Gills or heart are organs not tissue, they are composed of tissues (but not one tissue). Please correct terminology in the whole manuscript.

Reviewer 1 ·

Basic reporting

The manuscript is in conformance with the requests of PeerJ. The English is clear and easy to follow.
The introduction guides to the aim of the work which is clearly stated.

Experimental design

The Submission corresponds to the scope of the Journal. The Research question is defined and it is outlined, that a knowledge gab is filled with this work.
Some questions concerning the methods are indicated in the comments to the author part.

Validity of the findings

Data on which the conclusions are based on, are given in the result part and refers to the overall aim of the work.

Additional comments

The manuscript addresses an important point for the evaluation of organ samples by histological methods. Indeed the data provided are very helpful for histopathologists involved in fish pathology.
There are several questions which should be considered to make the manuscript more clear.

1. Two species were included in the study but no reference is given to possible differences between the two species throughout the paper. Further, it is not clear why two species were considered.
2. There is no indication on possible differences in samples from the same treatment and same timepoint
3. Numbers of samples not really clear: in abstract 39 fish or populations? But in text 30 samples. Not clear whether 30 animals per species or in total.
4. In the discussion organs other than gills, skin, intestinal tract and pancreas are not really considered. Some interpretation on findings of this organs would be helpful, in particular also in the conclusion part

Specific points:

Species names:
“Lepomis macrochirus” not “Lepomis macrohirus” (wrong troughout)
“Pomoxis annularis” not “Pomoxis anularis” (wrong troughout)

Materials and Methods
Line 66: Not really clear whether 30 animals of each species have been included. If not, how was the distribution of the species with regards to the sampling methods and points?
Line 76: Thawing process in bag or fish removed from bag?
Line 119: What was seen in other organs (spleen, liver and kidney)?
Line 140: Liver changes?
Line 169-170: Something missing in sentence
Line 173: Changes in brain?
Lines 211-212: Repetitive to lines 52-55 in introduction
Line 215: Were own results in agreement with these previously described findings i.e. link between literature and own results missing.
Line 235: Reference to own study missing
Line 262: Recommendations for organs other than gill and intestinal tract

Reviewer 2 ·

Basic reporting

Discussed under General Comments

Experimental design

Discussed under General Comments

Validity of the findings

Discussed under General Comments

Additional comments

This simple, straightforward, and informative study involves the examination of histologic sections of multiple fish tissue types for the relative degree of autolytic change following various time periods and holding conditions post sacrifice. Although the results were generally as anticipated, the documentation of postmortem artifacts under different handling conditions represents a valuable contribution to the fish pathology literature.

This paper has a number of positive attributes, including: the histologic processing and microscopic examination of eleven different tissue types; the use of a standardized semi-quantitative scoring system for assessing the relative amounts of autolysis per tissue specimen; inclusion of sufficient references; and very good quality photomicrographic figure images.

On the negative side, the experimental design for this experiment was less than optimal, because several variables, such as fish size, species, and sex, were not adequately controlled. This could be significant, especially in terms of fish size, because there were only three fish per handling method and time period. For example, the authors appropriately list fish size as an important influential factor for postmortem autolysis, and yet they provide no way for the reader to determine whether the different groups varied significantly by weight. Also, for the results to be more universally applicable, one would have expected other types of fishes (e.g., a small tropical aquarium species, and/or a marine fish) to have been tested in addition to these Centrarchids. The stated reason for the species selection in this case, which was that these fish were destined to be sacrificed in any event, is obviously commendable as opposed to terminating fish solely for experimental purposes. However, to some extent, the current study design does limit the degree to which the conclusions can be reasonably extrapolated to other fish species and sizes. This limitation should be stated in the Discussion or Conclusions section. It also would have been interesting to determine whether excised tissues would have fared better or worse than fish saved whole, but that former model was not investigated.

The focus of this manuscript appears to involve primarily diagnostic service type specimens involving display or pet fishes, which is fine. However, at some point in the mansucript (probably the discussion or conclusion) it is suggested that the authors emphasize that prospective research studies should not be designed to incorporate periods of postmortem delay prior to fixation. Even field collection studies should employ onboard or shore necropsy procedures whenever possible. Additionally, although formalin is one of the most widely available, versatile, and inexpensive fixatives, anectdotal and empirical evidence suggests that fixatives such as modified Davidsons or Bouins produce superior results with fewer artifacts for some fish tissue types, which can help the pathologist to identify subtle types of changes that may be more important for research specimens.

This manuscript is readily understandable, however the writing style is often awkward (the opening sentence of the abstract, for example), and there are numerous grammatical errors, most of which I did not take time to edit.

TITLE
The title is appropriate, although I prefer “histopathologic” to “histologic” when the procedures include slide examinations by a pathologist. Your choice.

ABSTRACT
Lines 8-11: From the abstract, one would assume that the two species of fish were assayed independently, and not pooled as appears to have been the case. This should probably be made more clear here and in the Introduction.

Line 11 (and line 62): Suggest change “value” to “quality”.

Line 11: The abstract says a total of 39 fish were used, but later in the paper it says 30. Which is correct?

Lines 11-12: Suggest change to “A total of 39 randomly selected mixed sex healthy bluegill… (I assume there weren’t 39 populations)

Lines 13-14: Suggest change to “Histologic specimens were evaluated by light microscopy”.

INTRODUCTION
Lines 59-62: Suggest change to “The purpose of this study was to qualitatively identify and score histologic postmortem…” (you’re not “qualifying” the changes, and you’re not really quantifying them either)

MATERIALS AND METHODS
Line 67: Strongly suggest including the range of fish weights in addition to the mean. It would also be a good idea to analyze the group weights statistically for significant differences.

Line 81: Suggest including the manufacturer of the NBF (or if you made it yourself, include a formula reference), and you should state that the same batch of NBF was used for all fish.

Line 88: You do not state what these percentages were percentages of. Probably not affected tissue area, so I assume they indicated percent difference from control or percent difference from what the pathologist considered to be normal. Please make this clear in the text.

Lines 88-89: Please indicate how many people scored the tissues, and whether the scorers were aware of the type of postmortem treatment associated with each fish.

Lines 95-96: Suggest change to “Criteria for autolysis comprised the following factors:” Suggest omitting the rest of the sentence, and instead insert this text as a separate sentence following the list “The overall severity of autolysis was based on the number of factors present and the magnitude of change in the affected cells/tissue.”

Line 102: The meaning of the parenthetical comment “(Note: lack of was not a limiting factor)” is not clear to me and perhaps should be rephrased or removed if not necessary.

RESULTS
Line 113: I’m not familiar with the incidental gill lesion of “occasional empty lacunae”. To my understanding, lacunae is an uncommonly used term for lamellar capillaries. What is the significance of empty lacunae? Please clarify this somehow in the text, or remove it.

Line 114: I suspect the phrase in this sentence should be “lamellar fusion” not “laminar fusion”? I’m also not familiar with “basal cell” hyperplasia of the secondary lamellae. Are you referring to hyperplasia of lamellar epithelial (pavement) cells at interlamellar sulci?

Line 132: I suspect you mean “mild amount”?

DISCUSSION and CONCLUSIONS
The discussion and conclusion sections seem appropriate and well thought out. The aforementioned experimental design limitations should be included as caveats.

TABLES
Table 1: This table is satisfactory as is, although it would be more intuitive and slightly easier to read if the type of storage method for each time point was simply listed as text in a column to the right of the “hours” column, as opposed to being represented by staggered X labels.

Table 2: For brain at 48h Z, I suspect “M to M” probably means mild to moderate, but as this could also mean minimal to mild it should be clarified (also for consistency).

SUPPLEMENTAL INFORMATION
The authors have furnished three documents as supplemental information. I don’t see the value of any of this information to the reader, even in supplemental form. Much of it is redundant with the text, and I would not include it. Far more valuable to include would be the individual fish weights, species, and sexes, and the individual tissue scores.

·

Basic reporting

No comments.

Experimental design

No comments.

Validity of the findings

As the authors state, subject generally known based on human and veterinary pathology. This work comes however as a very useful contribution especially but not restricted to fish histopathologists, due the rigorous and meticulous comparison, based on a simple but realistic and well-designed experiment.

Additional comments

-Easy to read and well written work.
-Suggest to use terminology for gills as "filament" and "lamellae" (rather than secondary lamellae), following Ferguson (2006), Systemic Pathology of Fish.
-Check line 169, double verb (were-was).

---

## Round 0.2 · Minor Revisions

· Academic Editor

Minor Revisions

Thank you for sending the revised manuscript and making corrections. It appears that you have not noticed a number of comments from reviewer 2 and these are not addressed. In particular:

>"However, to some extent, the current study design does limit the degree to which the conclusions can be reasonably extrapolated to other fish species and sizes. This limitation should be stated in the Discussion or Conclusions section. It also would have been interesting to determine whether excised tissues would have fared better or worse than fish saved whole, but that former model was not investigated."

Please state the limitation in the Discussion and Conclusion as requested.

While I understand that the authors cannot go back to their samples so that they cannot get individual or range of weights or species compositions, these are other limitations that should be acknowledged in the Discussion.

The same reviewers makes another statement which remains unanswered:

>"However, at some point in the mansucript (probably the discussion or conclusion) it is suggested that the authors emphasize that prospective research studies should not be designed to incorporate periods of postmortem delay prior to fixation. Even field collection studies should employ onboard or shore necropsy procedures whenever possible. Additionally, although formalin is one of the most widely available, versatile, and inexpensive fixatives, anectdotal and empirical evidence suggests that fixatives such as modified Davidsons or Bouins produce superior results with fewer artifacts for some fish tissue types, which can help the pathologist to identify subtle types of changes that may be more important for research specimens."

Please make sure this point is included in your Discussion.

And another unanswered point:

>"This manuscript is readily understandable, however the writing style is often awkward (the opening sentence of the abstract, for example), and there are numerous grammatical errors, most of which I did not take time to edit."

Please correct your manuscript.

I noticed that it is stated that the fish were "randomly" selected. Please explain what random selection procedures were used, I suspect that you mean "haphazardly" and not "randomly".

The word "tissue" is still used incorrectly - for example in Materials and Methods "Tissues from all fish were harvested in the following order: gills, upper and lower intestines, spleen, liver, stomach, gonad, posterior kidney, anterior kidney, heart, skin with underlying muscle, and brain. All tissue samples " - organs not tissues are listed. Please check definition of "tissue" and use correct terminology. Please correct throughout the whole manuscript.

---

## Round 0.3 · Minor Revisions

· Academic Editor

Minor Revisions

Dear author,

Thank you for making most of the revisions.

There are still a few things needing your attention, mostly related to editing of the paper. For example:

- line 71 - the word "randomly" is used - did you really apply random selection procedures or was it haphazard or you simply made sure that you had some males and females in your sample?
-line 341 - "Unfortunately, for logistical reasons, exact weights for each fish in this study was not recorded." - "weights' - plural so the verb should be in plural as well ("were' not "was")

Please carefully proof read your manuscript before the resubmission

---

## Round 0.4 · accepted · Accept

· Academic Editor

Accept

Thank you for making the changes.